# Immune-Activated B Cells Are Dominant in Prostate Cancer

**DOI:** 10.3390/cancers15030920

**Published:** 2023-02-01

**Authors:** Aws Saudi, Viqar Banday, A. Ali Zirakzadeh, Martin Selinger, Jon Forsberg, Martin Holmbom, Johan Henriksson, Mauritz Waldén, Farhood Alamdari, Firas Aljabery, Ola Winqvist, Amir Sherif

**Affiliations:** 1Department of Urology, Medical Faculty, Linköping University, 581 85 Linköping, Sweden; 2Department of Clinical and Experimental Medicine, Medical Faculty, Linköping University, 581 85 Linköping, Sweden; 3Department of Surgical and Perioperative Sciences, Urology and Andrology, Umea University, 901 85 Umea, Sweden; 4Department of Clinical Microbiology, Immunology, Umea University, 901 85 Umeå, Sweden; 5Immuneed AB, 753 41 Uppsala, Sweden; 6The Laboratory for Molecular Infection Medicine Sweden (MIMS), 901 87 Umeå, Sweden; 7Department of Molecular Biology, Umeå Centre for Microbial Research, 6K and 6L, Umeå University, 901 87 Umeå, Sweden; 8Department of Urology, Central Hospital of Karlstad, 652 30 Karlstad, Sweden; 9Department of Urology, Västmanland Hospital, 721 89 Västerås, Sweden; 10ABClabs, BioClinicum, Campus Solna, 171 76 Stockholm, Sweden

**Keywords:** prostatic neoplasms, B cells, T cells, sentinel lymph node biopsy

## Abstract

**Simple Summary:**

Contrary to the common belief that prostate cancer is an immune desert, our study shows tumor-associated B-cell responses in prostate cancer. We demonstrate mature and activated phenotypes of B cells with an increased frequency of effector plasmablasts in tumor-draining sentinel lymph nodes. These findings indicate a B-cell-specific antitumor immune response, emphasizing the importance of further trials targeting B cells in prostate cancer immunotherapy.

**Abstract:**

B cells are multifaceted immune cells responding robustly during immune surveillance against tumor antigens by presentation to T cells and switched immunoglobulin production. However, B cells are unstudied in prostate cancer (PCa). We used flow cytometry to analyze B-cell subpopulations in peripheral blood and lymph nodes from intermediate–high risk PCa patients. B-cell subpopulations were related to clinicopathological factors. B-cell-receptor single-cell sequencing and VDJ analysis identified clonal B-cell expansion in blood and lymph nodes. Pathological staging was pT2 in 16%, pT3a in 48%, and pT3b in 36%. Lymph node metastases occurred in 5/25 patients (20%). Compared to healthy donors, the peripheral blood CD19^+^ B-cell compartment was significantly decreased in PCa patients and dominated by naïve B cells. The nodal B-cell compartment had significantly increased fractions of CD19^+^ B cells and switched memory B cells. Plasmablasts were observed in tumor-draining sentinel lymph nodes (SNs). VDJ analysis revealed clonal expansion in lymph nodes. Thus, activated B cells are increased in SNs from PCa patients. The increased fraction of switched memory cells and plasmablasts together with the presence of clonally expanded B cells indicate tumor-specific T-cell-dependent responses from B cells, supporting an important role for B cells in the protection against tumors.

## 1. Introduction

Prostate cancer is still the second most commonly diagnosed cancer in men and the fifth cause of cancer-associated death among men worldwide [1]. Unlike other solid tumors, PCa has for a long time been considered an immunologically inert cancer because of the low tumor mutation burden with consequent low neoantigen expression [2]. However, PCa presents with a high number of mutations in DNA-damage and -repair genes [3,4,5], supporting the idea that prostate cancer is maybe immune-sensitive [6]. Focus and attention has been on T cells [7,8,9,10], promoted by using antibodies against immune checkpoint inhibitors (ICIs) which revolutionized cancer immunotherapy [11], but the effect of ICIs in the treatment of PCa [12] is less robust compared to other tumors. In contrast, the role of B cells has been overlooked until recently [13,14,15]. The growing findings of negative tumor modulation by B-cells with favorable prognosis in some human cancers [16] support the targeting of B cells in cancer immunotherapy. However, the role of B cells in the PCa microenvironment is still unclear.

B cells are multifaceted and can participate in immune responses through different mechanisms. Acting as potent antigen-presenting cells (APC), CD40 ligand-activated B cells induce antigen-specific CD8^+^ CTL cell and CD4^+^ T cell reactions against cancer cells [17]. Antibodies produced by B cells bind tumor antigens and further aid tumor cell eradication through antibody-dependent cell-mediated cytotoxicity (ADCC) or by the antibody-mediated activation of the complement system [18,19]. Furthermore, tumor-infiltrating B cells produce cytokines that stimulate the formation of tertiary lymphoid structures [20,21], which facilitate the formation of CD4 memory T cells [22] and proliferation of activated CD8^+^ T cells through CD27–CD70 interactions [23].

In mouse PCa models, immunosuppressive B cells support tumor progression [24] and promote the castration resistance phase of PCa, by activating IKK-alpha, STAT3, and BM1 [25]. Moreover, the elimination of immunosuppressive plasma cells that express IgA, IL-10, and PD-L1 support the CD8^+^ T-cell-dependent antitumor response associated with chemotherapy [26]. In humans, B cells have been documented in PCa with higher numbers being present in malignant than in benign prostate tissue [27]. Treatment with monoclonal antibodies against CD20 (rituximab) reduces tumor-infiltrating B and T lymphocytes, supporting the interaction between these two cells in PCa. The frequency of IL-10^+^CD19^+^ B cells correlates with the progressive stage of PCa [28]. It is important to distinguish the immunosuppressive from the immune-supportive B-cell phenotypes, since this distinction is not very clear in PCa and may lie behind the poor reputation of B cells.

Lymph nodes (LN) are important secondary lymphoid organs and are composed of different immune cells with a major role on the tumor host response. Tumor-draining lymph nodes, in particular, even called sentinel nodes (SN), act as a mediator of cancer cells leading to distant organ metastases [29]. Studies in animal models show 30–35% of cells in SNs consist of B cells [30], and B cells in SNs express regulatory characteristics and inhibit antitumor response [31,32]. In humans, PCa metastasis to pelvic lymph nodes promotes inhibitory immune responses [33]. In previous work, our group has demonstrated that B cells from SN in patients with cancer are activated and show clonal expansion specifically against the tumor [34], as well as that B cells in SN activate CD4^+^ T-cell-mediated anti-tumor responses. It is unclear whether the proximity of SN to the primary tumor and antigen exposure cause the SN microenvironment to become immunosuppressive, favoring tumor metastases or immune-protectiveness against tumor progression.

Little is known about B cells in human prostate cancer, particularly in pelvic lymph nodes. It is therefore of great interest to study different subgroups of B cells in PCa. We here make a global attempt to study B cells from all relevant locations simultaneously—peripheral blood, tumor draining, and non-draining lymph nodes from the same patient with intermediate-to-high-risk PCa, making our material collection unique.

## 2. Materials and Methods

### 2.1. Subjects and Sentinel Node Detection

We analyzed 81 tumor-draining (SN) and 52 non-draining lymph nodes (N-SN) collected from 25 patients with PCa which were included prospectively. The mean age of the 25 patients was 66.1 years (range 50–74 years). Pathological staging was pT2 in 4/25 (16%) patients, pT3a in 12/25 (48%), and pT3b in 9/25 (36%) patients. Lymph node metastases were found in 5/25 patients (20%). Additional clinical characteristics of the patients (tumor stage, Gleason score, prostate serum antigen (PSA), and lymph node count) are summarized in Table 1.

Peripheral blood was collected before surgery. Robot-assisted radical prostatectomy and extended pelvic lymphadenectomy was performed. Tumor-draining lymph nodes were identified and isolated separately from non-draining lymph nodes by two techniques. In 20 patients, trans-rectal ultrasound-guided radioactive 99mTechnetium Nano colloid was injected in the prostate one hour before surgery. Geiger meter was used to identify lymph nodes intraoperatively, and positive lymph nodes were designated as SN, whereas in 5 patients, Indocyanine-Green-guided lymph node dissection was performed by trans-rectal ultrasound-guided injection of 8 mL Indocyanine Green 5 mg/mL in the prostate, 45–60 min before surgery. Green-stained lymph nodes were isolated perioperatively and designated as SN, followed by extended pelvic lymph node dissection for isolation of N-SN.

After surgery, all SN and N-SN were divided in the middle; one half was sent for histopathological analysis and the other half for cell isolation and flow cytometry. Blood and lymph nodes were preserved in ice bags and transported freshly on the same day for analysis.

### 2.2. Cell Preparation

Specimens were transported from operation room to the laboratory at 4 °C and processed within 24 h. Peripheral blood mononuclear cells (PBMC) were isolated using density gradient centrifugation using Ficoll-Paque PLUS (GE Healthcare, Chicago, IL, USA) and SepMate tubes (StemCell Technologies, Vancouver, BC, Canada). Cells were isolated from LNs by gently pressing through a 70 µm cell strainer and suspended in PBS supplemented with 3% FBS (Gibco, Waltham, MA, USA).

### 2.3. Flow Cytometry

Cells were washed in a FACS buffer containing PBS, 2% FBS and 0.05% NaN_3_. Cells were stained as previously described [34]. Briefly, post wash, the cells were stained for surface markers with the following antibodies: APC-Cy7/Alexa fluor 700 conjugated anti-human CD19, FITC conjugated anti-human IgD, PE-Cy7/APC conjugated anti-human CD38, PE-Cy5/BV605 conjugated anti-human IgM, PE/BUV395 conjugated anti-human CD27, APC/PE-Cy7 conjugated anti-human IgK, BV421 or Alexa fluor 647 conjugated anti-human IgL, BV421 conjugated anti-human IgG, and PE conjugated anti-human IgA (all purchased from BD Biosciences, San Jose, CA, USA). Live cells were detected using fixable LIVE/DEAD stain (Thermo Scientific, Waltham, MA, USA). Samples were analyzed on a FACS LSRII and Bio-Rad ZE5 using the FCS express software.

### 2.4. VDJ Sequencing and Clonal Analysis

CD19^+^ B cells from tumor-draining lymph nodes (S), non-draining lymph nodes (N), and peripheral blood (P) were positively isolated using a Dynabeads Pan B cell kit (Thermo Scientific, Waltham, MA, USA #11143D). Isolated B cells were washed and counted. Nucleospin XS kit (Macherey-Nagel, Dueren, Germany #740902.50) was used for total RNA isolation according to the manufacturer’s instructions.

Quality and integrity of total RNA isolated from isolated B lymphocytes were first assessed using the Agilent 2100 Bionanalyzer. Subsequently, 1–10 ng of total RNA was used for the generation of IgM/IgG heavy and light chain libraries of V(D)J variable regions of BCR transcripts employing the SMARTer Human BCR IgG IgM H/K/L Profiling Kit (Takara Bio, Shiga, Japan #634466). Purified and size-selected libraries were validated using the Agilent 2100 Bionanalyzer.

Pooled libraries (15 pM) were sequenced on an Illumina MiSeq platform using the 600-cycle MiSeq Reagent Kit v3 (Illumina, San Diego, CA, USA #MS-102-3003) with paired-end, 2 × 300 bp reads. A 10% PhiX control spike-in was used in order to increase the nucleotide diversity.

Raw FASTQ files were firstly processed using Cogent NGS Immune Profiler Software v 1.0 (Takara Bio). Resulting data containing V(D)J mapping information and statistics served as an input for further BCR profiling analyses and visualization using the immunArch R package version 0.6.8 [35].

### 2.5. Statistics

Graphs represent mean ± SEM. Statistical analysis was performed by using unpaired Student’s t-tests or ANOVA multiple comparison test; *p* < 0.05 were considered as statistically significant.

## 3. Results

### 3.1. B-Cell Distribution in Blood and Lymph Nodes from Patients with PCa

To investigate the distribution of B cells in patients with intermediate- and high-risk prostate cancer, we utilized blood, SN, and N-SNs harvested during surgery. We used the validated and previously used Freiburg panel to explore B-cell populations in the PCa patients [34]. Using a panel of cell-surface markers, the CD19^+^ B cells were categorized into six subpopulations: naive B cells (IgD^+^CD27^−^), marginal-zone-like/natural-effector B cells (IgD^+^CD27^+^), class-switched memory B cells (IgD^−^CD27^+^), transitional B cells (IgM^++^CD38^++^), and plasmablasts (IgM^−^CD38^+++^). A representative gating strategy utilizing the Freiburg panel is shown in Figure 1a,b.

When analyzing the CD19^+^ B cells, we observed a significant accumulation of B cells in LN compared to blood as expected (Figure 2a). The general distribution of B-cell subpopulations is different in blood and in LN. The proportion of naïve cells and transitional cells dominate in blood (Figure 2b,c) whereas switched memory and plasmablasts accumulate in LN (Figure 2d,e). No differences were found in the marginel zone compartment (Figure 2f). When investigating the light chain usage, we found signs of clonal expansion within lymph nodes and blood samples skewing towards Igλ dominance (Igλ/Igκ > 0.7) (Figure 2g). B cells positive for the early activation marker CD69 and the inducible costimulatory molecule CD86 were increased in lymph nodes, whereas, interestingly, the constitutively expressed costimulatory molecule CD80 was decreased on B cells from lymph nodes (Appendix A), suggesting ongoing activation in lymph nodes.

### 3.2. Decreased Fraction of Circulating B Cells in Prostate Cancer Patients

To further study peripheral B-cell subpopulations in the PBMC of PCa patients (P-PBMC), we compared 17 patient samples with samples from 10 healthy blood donors (H-PBMC) as previously described [34], noticing that the distribution of B-cell subpopulations in our healthy donor cohort are within the normal reference range of the Freiburg panel [36].

However, we observed a significantly decreased fraction of CD19^+^ B cells in P-PBMC compared to H-PBMC (*p* < 0.001) (Figure 3a), and significantly smaller plasmablast and MZ fractions in P-PBMC compared to H-PBMC (Figure 3e,f) (*p* < 0.01 and *p* < 0.05, respectively). Consequently, significantly increased fractions of transitional B cells were observed in P-PBMC compared to H-PBMC (Figure 3b, *p* < 0.05). No differences were noticed among switched memory or naive B cells (Figure 3c,d). When analyzing the Igλ/Igκ ratio, we observed the expected normal Igκ dominance (ratio 0.7) in H-PBMC (Figure 3g). Interestingly, in P-PBMC, we observed signs of Igλ-dominant clonal expansions since the majority of P-PBMC present has an Igλ/Igκ ratio above the 0.7 threshold, although not reaching significance (Figure 3g). However, at the individual level, we find the increased usage of the Igλ light chain in six out of 13 samples, where the highest Igλ/Igκ ratio was found to be as high as 1.7. The very early cell activation marker CD69 was decreased in P-PBMCs although not reaching significant levels (Appendix A). Thus, there are fewer activated B cells, plasmablasts, and marginal-zone B cells in blood from PCa patients compared to healthy controls.

### 3.3. Clinical Comparison in Blood and Lymph Nodes

Furthermore, we analyzed the relation of pre-op PSA, Gleason score (Gs), local stage of tumor, and the presence of lymph node metastasis with peripheral CD19^+^ B cells and B-cell subpopulations. We found significantly decreased amounts of CD19^+^ B cells in the P-PBMC in patients in the advanced stage (*p* < 0.05) (Figure 4c, CD19 panel) and a general trend for decreased CD19^+^ cells with PSA ≥ 20 and Gs ≥ 7, although not reaching significant levels (Figure 4a,b CD19). Interestingly, we found significantly increased CD19^+^ B cells in the P-PBMC in patients with lymph node metastasis compared to those with no lymph node metastasis (*p* < 0.01) (Figure 4d, CD19 panel).

When analyzing lymph node B cells, we observed a significantly increased fraction of CD19^+^ B cells in samples from patients with advanced Gs (*p* < 0.001) and the stage with a concomitant decrease in transitional and naïve B-cell numbers (Figure 4b,c). Significantly decreased plasmablast numbers were observed in the advanced clinical characteristics stage (*p* < 0.001) and metastasis (*p* < 0.01) while not reaching significant levels in Gs (*p* < 0.0565) (Figure 4b–d). Nodes from both the lower and advanced clinical characteristics stage showed clonal expansion with some demonstrating the normal kappa dominance whereas the majority displayed a dominant usage of the lambda light chain. However, majority of lymph nodes from advanced clinical parameters were above the 0.7 threshold (Figure 4c).

### 3.4. B Cells and IgG-Producing Plasmablasts Accumulate in SN

To analyze if tumor drainage affected B-cell populations in lymph nodes, we compared B cells from SN, N-SN, and peripheral blood using the Freiburg panel. We found a significantly increased fraction of CD19^+^ in SN compared to N-SN (*p* < 0.05) (Figure 5a). Compared to blood, significantly lower numbers of transitional cells and naïve cells were observed in the lymph nodes (*p* < 0.05) (Figure 5b,c). In addition, an increased fraction of plasmablasts was observed in SN (Figure 5e–i) (*p* < 0.05). When analyzing the heavy chain usage in plasmablasts, we observed significantly increased IgG usage compared to IgA (*p* < 0.01) (Figure 5i). In contrast, the fraction of IgA^+^ switched memory B cells was found to be increased in SN (*p* < 0.05) (Figure 5h). When investigating the relation between MZ and plasmablasts, we found a constant MZ dominance in N-SN, whereas in SN, we found a plasmablast dominance in six samples, suggesting activation in response to the tumor (Figure 5j,k). Furthermore, when analyzing the Igλ/Igκ ratio, we again observed a lambda light chain dominance in SN and N-SN compared to blood, indicating the clonal expansion of B cells in SN and N-SN (Figure 5g) (*p* < 0.05).

### 3.5. VDJ Analysis

The findings of activation, maturation, and switched light chain usage suggest the clonal expansion of tumor-antigen-specific B cells. To verify the clonal expansion of B cells in lymph nodes, we performed B-cell receptor (BCR) profiling of transcripts encoding V(D)J-variable regions of IgG/IgM heavy and light (Igκ, Igλ) chains. BCR profiling analysis involved SN (S), N-SN (N), and peripheral blood (P) samples from three patients. In total, 44,640, 54,748, and 19,499 unique clonotypes were identified in patients 1, 2, and 3, respectively (Appendix A). The highest diversity in all three patients was observed in peripheral blood samples for heavy chains of the IgM isotype. Nevertheless, a high degree of diversity for IgM isotypes was determined across all tissues (Appendix A). An analysis of the relative clonotype abundance (i.e., the proportion of BCR repertoire occupied by clonal groups with specific abundances) further verified the phenomenon of high clonotype variability in peripheral blood when compared to SN and N-SN (Figure 6a). Furthermore, the relative clonotype abundance data point towards strong clonal expansion in SN in patient 1 and initialized clonal expansion in patient 4 (Figure 6a).

Interestingly, the high overlap count of BCR repertoires between the analyzed patients was documented for light chains (Igκ and Igλ), but not for heavy chains (Figure 6b, Appendix A). The high rate of clonotype overlap among analyzed patients led us to the question whether we can identify some of the hyperexpanded clonotypes as conserved among patients or tissues. For this purpose, the top five most abundant clonotypes in SN samples from all three patients were tracked in all datasets for Igκ and Igλ chains.

Figure 6c,d demonstrates that patient 1 (strong clonal expansion in SN) and patient 3 (initial clonal expansion in SN) contribute with the highest proportion (13–28%) of the top five most abundant clonotypes; however, the hyperexpanded clonotypes in patient 1 were not determined as hyperexpanded in patient 3. On the other hand, the hyperexpanded clonotypes CQQYYSTPWTF (IGKV4.1-IGKJ1), and CQQYNSYPYTF (IGKV1.5-IGKJ2) identified in patient 1 (SN) were found to be conserved across all Igκ samples analyzed (Figure 6c, Appendix A). Interestingly, the hyperexpanded clonotypes identified for Igλ chains were found to be rather unique in the SN of patient 1, with clonotype CCSYAGSYTFEWVF (IGLV2.11-IGLJ3) as the most prominently hyperexpanded (Figure 6d, Appendix A). Thus, the hyperexpanded clonotypes and the conserved sequence usage found in the sentinel node draining the tumor suggest the antigen-specific expansion of B cells towards the tumor antigen.

## 4. Discussion

Here, we demonstrate, in a unique material derived from prostate cancer patients, the expansion, activation, and clonal expansion of B cells in tumor-draining sentinel lymph nodes. B cells from PCa were found to display a mature and activated phenotype with an increased frequency of effector plasmablasts in SNs. Together, our results demonstrate an anti-tumoral B-cell response in patients with prostate cancer, a tumor form usually regarded as non-immunological in contrast to urinary bladder cancer.

Studies comparing peripheral and lymphoid compartments in prostate cancer are missing. In this study, we describe B cells and their subpopulations in prostate cancer patients utilizing a unique extended material collection of blood, SN, and NSN sampled from the same patients allowing the simultaneous comparison of compartments. The Freiburg panel used in this study was originally designed for a comprehensive characterization of B-cell subpopulations in samples from patients with common variable immunodeficiency (36). Previously, we demonstrated the usefulness of the Freiburg panel to characterize B-cell subpopulations from patients with urinary bladder and colon cancer [34]. To the best of our knowledge, this is the first time B-cell subpopulations from simultaneous collected samples from peripheral blood and lymph nodes from intermediate-to-high-risk prostate cancer patients have been investigated.

As expected, comparing peripheral blood and lymph nodes, we observed a higher proportion of CD19^+^ B cells in lymph nodes. This difference in proportions was expected since B cells after maturation and activation home to secondary lymphoid organs. Accordingly, the majority of B cells in peripheral blood were of the transitional and naïve type (Figure 2), whereas plasmablast and switched memory cells dominated in lymph nodes. The Ig receptor is composed of heavy and light chains where initially, when the Ig is rearranged, a kappa light (Igk) chain is chosen in a 2:1 ratio compared to the lambda chain (Igλ). As part of affinity maturation or avoiding autoimmunity, the B cell may be instructed to change and rearrange to instead use the Igλ chain. The ratio between kappa and lambda can be used as an indication for the switching and clonal expansion of B-cell clones. When analyzing light chain usage, we observed variations in B cells from blood, whereas B cells from lymph nodes frequently were expressing the Igλ light chain (Figure 2g). In addition, there was an increased Igλ light chain usage in patient PBMCs (Figure 3g), suggesting clonal expansion and light chain switch in patients as a response to tumor antigens.

With indicative findings suggesting clonal expansion of tumor-reactive B-cell clones, we decided to analyze individual B-cell receptors (BCR) by sequencing. BCR profiling of V(D)J-variable regions demonstrated clonally expanded B cells accumulated predominantly in SN. This phenomenon was observed to be patient-specific (Figure 6a). Two out of the three investigated patients demonstrated clonal expansion of B cells, in line with unique and individual anti-tumoral immune responses. It will be interesting to follow the patients and correlate clonal B-cell expansion with clinical outcome. Despite the abundance of unique patient individual clonotypes that were diverse and rather patient-specific, we also found a high ratio of public clonotypes sharing the same IGHV gene present in both light chains of IgM and IgG isotypes (Figure 6b). Additionally, hyperexpanded clonotypes identified in patient 1 (clonal expansion in SN) were derived from the public clonotype repertoire in the case of Igκ, but not Igλ (Figure 6c,d). These findings suggest a common and conserved epitope activating B cells, likely representing an important tumor antigen. Activities have been initiated to try and elucidate the nature of the prostate cancer antigen, which could be a potential target for immunotherapy in the future.

Unexpectedly, we found a decreased proportion of CD19^+^ B cells in blood from PCa patients (Figure 3a). However, the decrease in CD19^+^ B cells was paralleled by an increased proportion of naïve CD19^+^ B cells. The changes found in the blood of PCa patients may thus be a direct consequence of the migration of activated cells to secondary lymphoid organs and a concomitant increase in the bone marrow output of naïve B cells in PCa patients. Supporting this notion, a significantly decreased number of effector plasmablasts in peripheral blood from patients suggests the migration of plasmablasts to other compartments such as the bone marrow, lymph nodes, or to the tumor (Figure 2d,e and Figure 3e). In addition, the decreased fraction of marginal-zone-like B cells in patients’ blood might reflect the rapid conversion to plasmablasts which transmigrate and leave the blood stream (Figure 3f). Thus, we found a redistribution of B-cell subpopulations in the blood from PCa patients suggesting a systemic effect of the tumor presence.

In prostate cancer, the PSA level, Gleason score, and local stage are vital for the prediction of the prognosis [37,38,39]. The clinical correlation between poor tumor characteristics (PSA > 20, locally advanced tumor or low-grade tumors), increased risk of metastasis, and worse prognosis are well-studied. However, the correlation of tumor characteristics and immune surveillance is not clear, particularly with regard to the role of B cells. Prostate tumors have increased infiltration of plasma cells and IgG expression [40], where increased plasma cell infiltration was associated with improved disease-free survival following radical prostatectomy. Treatment with Sipuleucel-T dendritic cells correlated with elevated serum levels of anti-tumor IgG and longer survival [41], suggesting that plasma cells may promote an antibody-dependent effect within the tumor microenvironment. In addition to producing soluble IgG that can activate the complement and induce antibody-dependent cellular cytotoxicity of tumor cells, B cells through their tumor-antigen-specific BCR provide antigen presentation to tumor-specific T cells. Thus, B cells have multiple roles and functions supported by our recent findings that tumor-infiltrating B cells also coexist with IgG and complement deposits in urinary bladder cancers [42].

We found correlations between B cells and tumor characteristics, PSA levels, tumor stage, and grade. First, we revealed a significantly increased fraction of CD19^+^ B cells in LN from T3 compared with T2 patients and with a worse Gleason score PCa but not for PSA level. In parallel, we found a decreased proportion of CD19^+^ B cells in the P-PBMC in patients with worse tumor characteristics and a higher lambda light chain index observed both in the blood and LN in worse tumor characteristics. Phenotyping of B cells reveal less transitional cells and naive cells in LN with worse tumor characteristics as well as less plasma blast in LN with worse characteristics, while the level of switched memory cells is less affected indicating that switched memory cells remain in the LN without further migration. Interestingly, patients with LN metastasis have significantly higher P-PBMC CD19^+^ B cells compared with non-metastasis patients. In addition, these patients have less plasmablasts in both blood and LN, however, not reaching significant levels in blood. It is important to be aware that tumor characteristics are dynamic and the patients progress to worse characteristics over time. It is tempting to speculate that a decrease in plasmablasts is a sign of exhaustion of B cells or a rapid migration to bone marrow. These findings are unique and show that a decrease in a potent B-cell immune response correlated to poor PCa tumor characteristics. The clinical impact of these findings needs to be studied in a long-term follow-up.

SNs are the first nodes that receive lymphatic drainage with tumor antigens and therefore the primary center for immune surveillance, where both T and B cells are primed [43]. However, the concept of the SN where metastases do not occur in other nodes if sentinel LNs are free of tumors remains controversial in PCa. B lymphocytes have many functions and roles. The most obvious function is the ability of B lymphocytes to produce soluble immunoglobulins to opsonize and activate the complement and antibody-dependent cellular toxicity (ADCC). However, by binding antigens to the cell-surface BCR on B lymphocytes, internalization, processing, and presentation antigen peptides in the HLA-class II pocket will be induced, resulting in the activation of specific TCRs on T lymphocytes. Moreover, internalized processed antigens may also be released as extracellular vesicles (EVs) with the capability of antigen presentation allowing distant crosstalk between B lymphocytes and T lymphocytes pioneered by Raposo et al. [44]. The immunological role of clonally expanded B lymphocytes, as demonstrated in this paper, and their mode of action(s) in prostate cancer needs to be further elucidated. In our previous study, we have demonstrated strong signs of T-cell-dependent B-cell responses in tumor-draining lymph nodes from melanoma patients [34]. In breast cancer, higher levels of B cells within SNs were shown to prevent future recurrence [45]. In our study, when we compared sentinel nodes to non-sentinel nodes from PCa patients, we observed a significant increase in CD19^+^ B cells and plasmablasts in sentinel nodes. This might reflect the increased differentiation of mature tumor-specific B cells before the migration to bone marrow. Although we did not evaluate B-cell homing factors such as CXCL13, any difference in expression of CXCL13 between SN and N-SN could be reflected in the proportions of B cells and plasmablasts populating the nodes. Differences were also noticed in the B cell isotypes in SN and N-SN. Whereas IgA-positive switched memory cells dominated, IgG^+^ plasmablasts were significantly increased in SN. This may reflect the functional aspects of the isotypes. IgA and IgG are two isotopes which have been shown to activate efficient antibody-dependent cell-mediated cytotoxicity/phagocytosis by a variety of mechanisms [46,47].

Although we have described the different B-cell populations in peripheral and nodal compartments, there are limitations to our study. First, our cohort of PCa patients is quite limited in number and the percentage of patients with metastasis is not big enough to draw firm conclusions. Secondly, although we compared thet peripheral blood of patients with healthy donors, we did not compare lymph nodes from non-cancer patients because of ethical considerations and the risk for complications. In addition, due to clinical limitations, we were not able to obtain enough tumor material for a description of TIL from the whole cohort. Finally, using two different techniques for the isolation of sentinel nodes might affect the validation of our material. In spite of our limitations, our results efficiently describe the variability of different B-cell subpopulations in PCa that could be observed even in a small population.

## 5. Conclusions

Our study demonstrates tumor-associated B-cell responses in PC, where the maturation of B cells in sentinel nodes and increased lambda light chain usage together with VDJ clonality suggest B-cell responses towards tumor antigens. Further follow-up studies are required to address the role of B cells in prognosis and as predictors of metastasis.

## Figures and Tables

**Figure 1 cancers-15-00920-f001:**
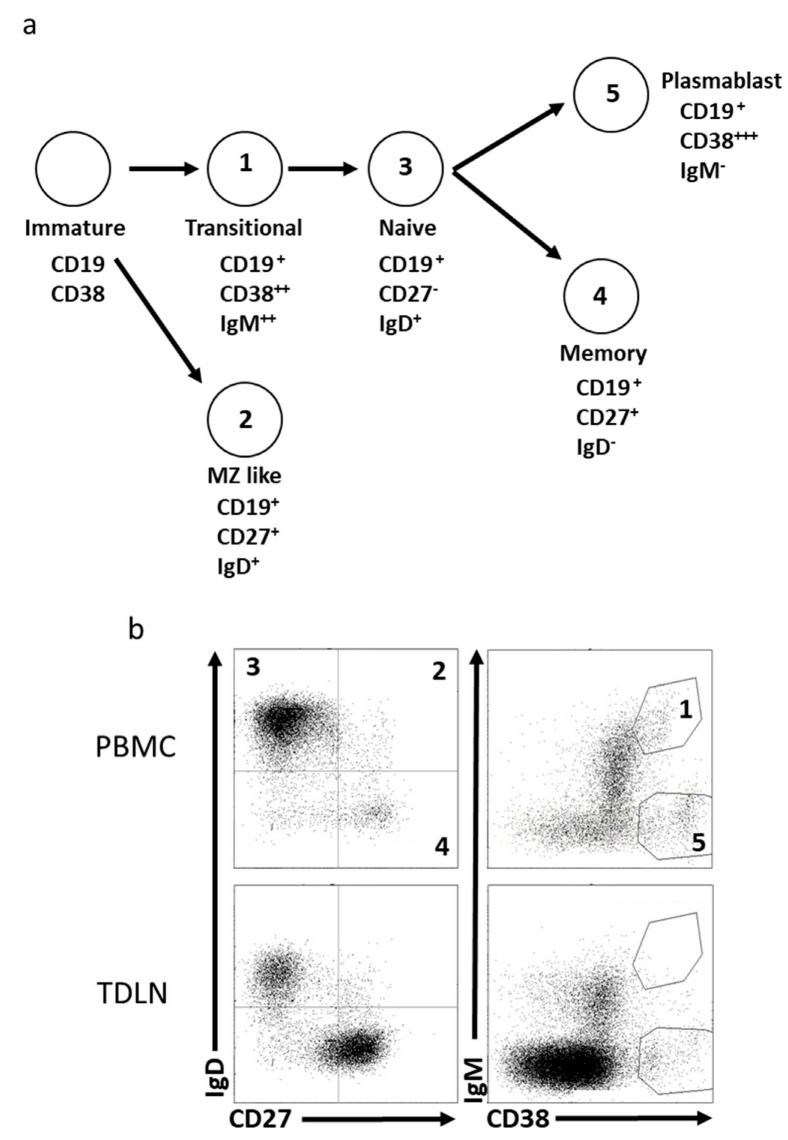
Representative gating strategy using the Freiburg panel. The B cells can be divided into five subpopulations based on expression of surface markers (**a**). Gating strategy revealing the five subpopulations. The numbers correspond to different subpopulations (**b**). All cells are gated on live CD19^+^ cells.

**Figure 2 cancers-15-00920-f002:**
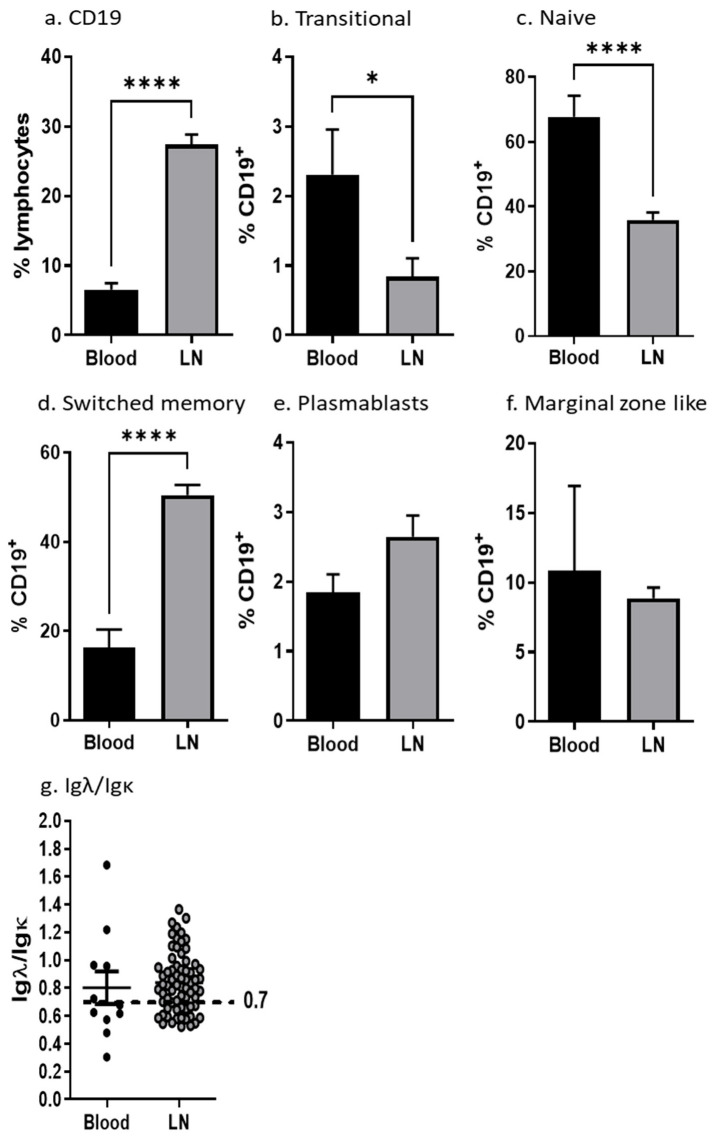
CD19+ B cell distribution in different cell compartments. Cells isolated from blood and lymph nodes were stained with surface markers and analyzed on a flow cytometer. (**a**–**f**) Mean percentages of CD19^+^ B cells in total lymphocytes and of the B-cell subpopulations in CD19^+^ lymphocytes are shown. (**g**) Igλ/Igκ isotype distribution was determined by flow cytometry. The proportion is calculated by dividing Igλ with Igκ percentages of CD19^+^ cells. (n blood = 13; n LN = 74). All error bars indicate SEM. Percentages of lymphocyte subpopulations were compared with Student’s two-tailed *t*-test. *p*: * < 0.05, **** < 0.0001.

**Figure 3 cancers-15-00920-f003:**
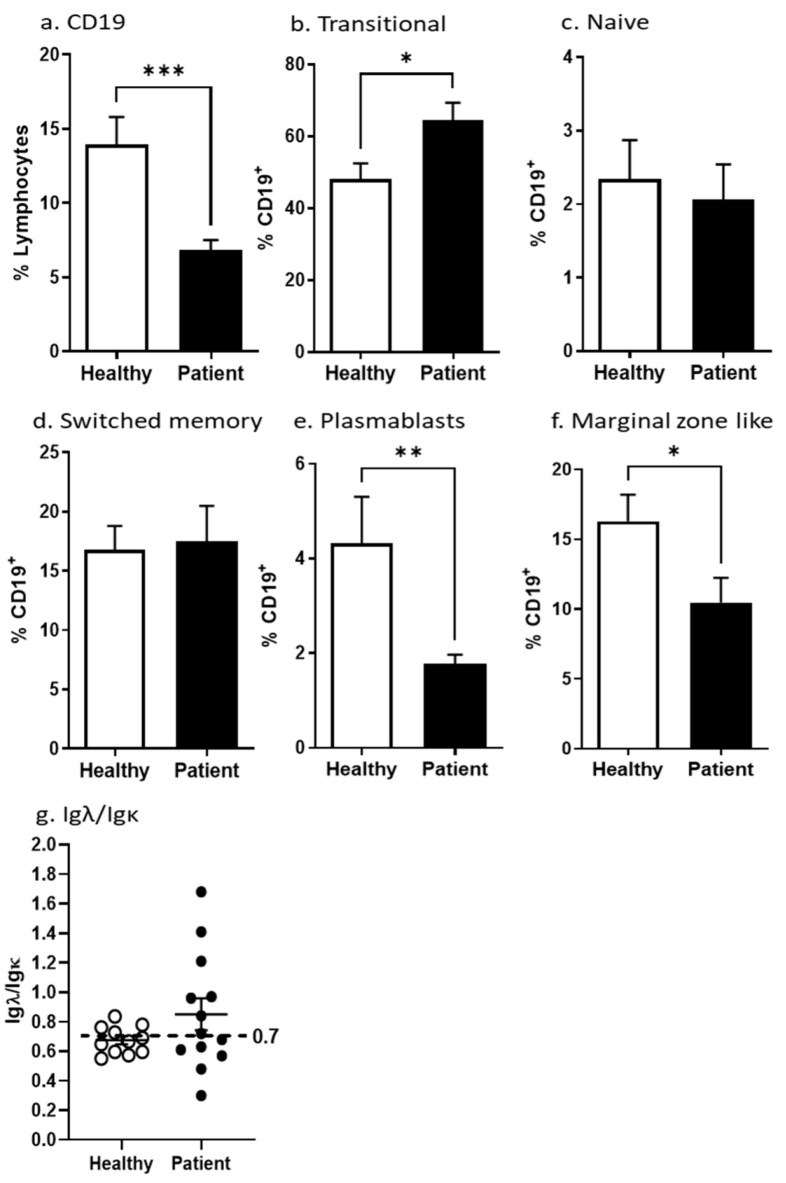
Peripheral B-cell compartment in prostate cancer patients and healthy donors. Buffy coat cells isolated from blood were stained and analyzed on a flow cytometer. Mean percentages of CD19^+^ B cells in total lymphocytes and of the B-cell subpopulations in CD19^+^ lymphocytes are shown. Igλ/Igκ isotype distribution was determined by flow cytometry. The proportion is calculated by dividing Igλ with Igκ percentages of CD19^+^ cells. All error bars indicate SEM. Percentages of lymphocyte subpopulations were compared with the Student two-tailed *t*-test. *p*: * ≤ 0.05, ** ≤ 0.01 *** ≤ 0.001.

**Figure 4 cancers-15-00920-f004:**
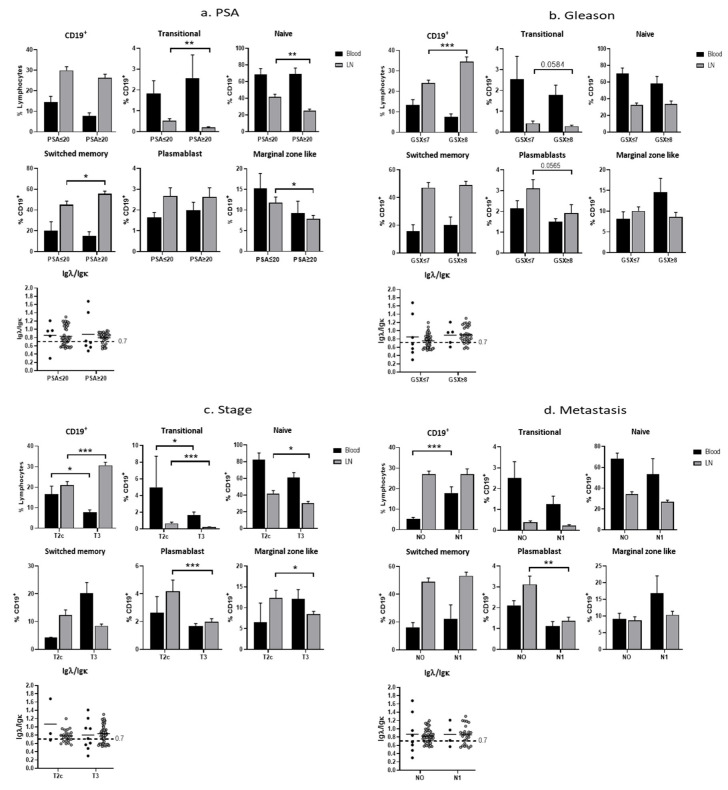
Correlation of clinical parameters and CD19 B cells in peripheral blood and lymph nodes of patients. Patients were stratified based on clinical parameters and analyzed for B-cell populations. Mean percentages of CD19^+^ B cells in total lymphocytes and of the B-cell subpopulations in CD19^+^ lymphocytes are shown. Dotted line marks the expected normal IgL/IgK ratio of 0.7. All error bars indicate SEM. Percentages of lymphocyte subpopulations were compared with the Student two-tailed *t*-test. *p*: * ≤ 0.05, ** ≤ 0.01 *** ≤ 0.001.

**Figure 5 cancers-15-00920-f005:**
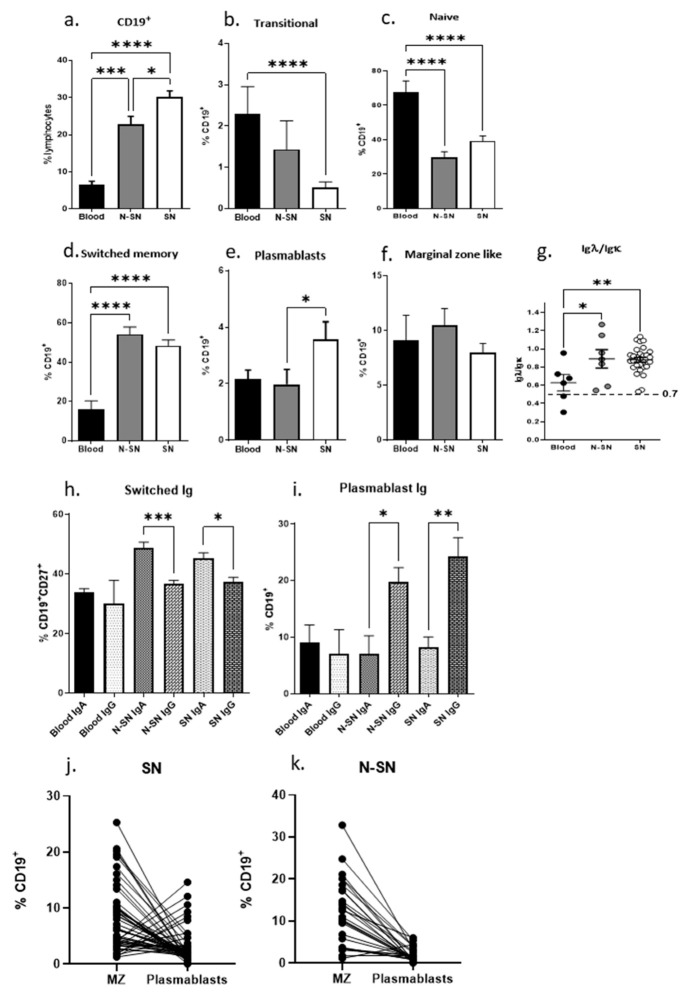
B-cell distribution in SN and N-SN. (**a**) Correlation of clinical parameters and CD19 B cells in lymph nodes of patients. Patients were stratified based on clinical parameters and analyzed for B-cell populations. (**b**,**c**) B cells stained for different isotypes. Mean percentages of CD19^+^ B cells in total lymphocytes and of the B-cell subpopulations in CD19^+^ lymphocytes are shown. All error bars indicate SEM. Percentages of lymphocyte subpopulations were compared with the Student two-tailed *t*-test. *p*: * ≤ 0.05, ** ≤ 0.01 *** ≤ 0.001 **** ≤ 0.0001.

**Figure 6 cancers-15-00920-f006:**
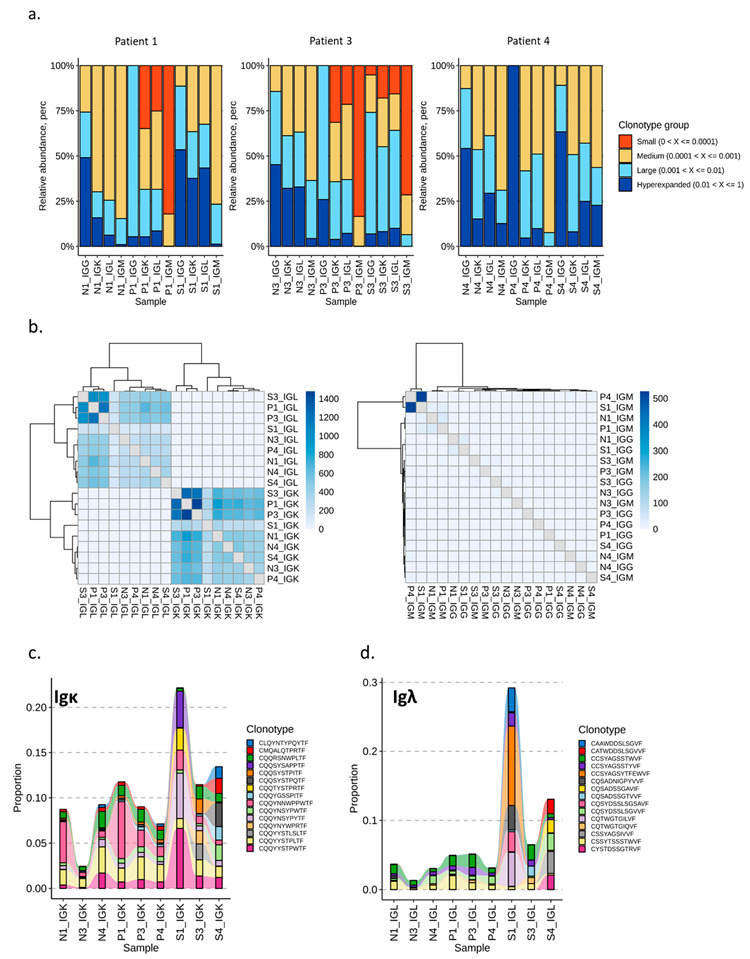
V(D)J sequencing and analysis. B cells were purified from N-SN (N), peripheral blood (P), and SN (S) from three patients (patient 1–N1, P1, S1–PSA: 9.9, Gleason Score: 7a, Stage: T3a, Age: 71; patient 3–N3, P3, S3- PSA: 8.6, Gleason Score: 9, Stage: T3a, Age: 78; patient 4–N4, P4, S4–PSA: 8.6, Gleason Score: 6, Stage: T2, Age: 67). Total RNA was isolated and subsequently used for cDNA synthesis and library preparation of CDR3 region of V(D)J segments using SMARTer Human BCR IgG IgM H/K/L Profiling Kit (Takara). The PCR product was sequenced (MiSeq Illumina platform) and analyzed using the Immunarch package. (**a**) Relative abundance for clonotypes identified in N-SN, peripheral blood, and SN. (**b**) Heatmap representing hierarchical clustering according to the number of public clonotypes across light (left) and heavy (right) chains of IgG and IgM isotypes. (**c**) Visualization of tracking the top 5 Igκ clonotypes identified in SN across all samples. (**d**) Visualization of tracking the top 5 Igλ clonotypes identified in SN across all samples.

**Table 1 cancers-15-00920-t001:** Intermediate-to-high-risk patient group. Stages and Gleason scores determined from pathology post-surgery.

n	Patients = 25 Donors = 10
Age (years)	66.1 (50–74)
PSA mean (range; µg/L)	17.2 (3–46)
Stage	pT2 = 4
	pT3a = 12
	pT3b = 9
Gleason Score	GS7a = 6
	GS7b = 6
	GS8 = 7
	GS9 = 6
pN1/pN0	pN1 = 5
	pN0 = 20

## Data Availability

Anonymous (coded) clinical patient data can, on reasonable request, be acquired from the last senior author. The BCR-sequencing data are available at ArrayExpress with accession #E-MTAB-11862. The R code for the analysis is available at Github: https://github.com/henriksson-lab/Immune-activated-B-cells-are-dominant-in-Prostate-cancer (accessed on 30 October 2022).

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
