# Peer review of "Immune-Activated B Cells Are Dominant in Prostate Cancer"

_cancers, 2023, doi:10.3390/cancers15030920_

Round 1
Reviewer 1 Report
In this study, the authors showed increased fractions of CD19+ B cells and switched memory B cells in lymph nodes in PCa patients. Plasmablasts were increased in tumor draining sentinel lymph nodes compared to non-draining lymph nodes. BCR-seq analysis indicated clonal expansion in lymph nodes. These findings support tumor specific T cell-dependent responses from B cells.
Below are some questions that the authors need to address:
- Please define abbreviations on first use in the manuscripts. For example, “PBMC” stands for “peripheral blood mononuclear cell”.
- In Line 212, it should be “ratio 0.7” instead of “ratio 0,7”.
- In Lines 218-220, the authors need more evidence to support the conclusion about “redistribution to secondary lymphoid organs or to tissues”.
- In Fig3. g, the dash line between 0.6 and 0.8 instead of between 0.4 and 0.6.
- In Line 284, the sample IDs are different from that in Fig. 6. Please be consistent.
- Could the authors do mass spec with the tumor samples from patient 1 to identify the tumor specific antigens? If so, by comparing the sequence of tumor-specific antigens and Igλ chains, the authors may be able to validate that the unique prominently hyperexpanded clonotype CCSYAGSYTFEWVF 306 (IGLV2.11-IGLJ3) was induced by tumor-specific antigens in patient 1.
Author Response
Below are some questions that the authors need to address:
- Please define abbreviations on first use in the manuscripts. For example, “PBMC” stands for “peripheral blood mononuclear cell”.
Reply: Abbreviations have now been explained in the text according to appearance.
- In Line 212, it should be “ratio 0.7” instead of “ratio 0,7”.
Reply: Decimal has been inserted instead of comma.
- In Lines 218-220, the authors need more evidence to support the conclusion about “redistribution to secondary lymphoid organs or to tissues”.
Reply: We agree with the reviewer and the sentence has been removed.
- In Fig3. g, the dash line between 0.6 and 0.8 instead of between 0.4 and 0.6.
Reply: The figure has been edited accordingly.
- In Line 284, the sample IDs are different from that in Fig. 6. Please be consistent.
Reply: The sample IDs should now be consistent throughout the manuscript.
- Could the authors do mass spec with the tumor samples from patient 1 to identify the tumor specific antigens? If so, by comparing the sequence of tumor-specific antigens and Igλ chains, the authors may be able to validate that the unique prominently hyperexpanded clonotype CCSYAGSYTFEWVF 306 (IGLV2.11-IGLJ3) was induced by tumor-specific antigens in patient 1.
Reply: We agree that identifying the target antigen is important in understanding response by immune cells and at the same time help in development of potential diagnostic and therapeutic biological products. As mentioned in lines 375-376, we have started patient inclusions for assessing protein expression in tumor material. The study is planned as a prospective study.
Below are some questions that the authors need to address:
- Please define abbreviations on first use in the manuscripts. For example, “PBMC” stands for “peripheral blood mononuclear cell”.
Reply: Abbreviations have now been explained in the text according to appearance.
- In Line 212, it should be “ratio 0.7” instead of “ratio 0,7”.
Reply: Decimal has been inserted instead of comma.
- In Lines 218-220, the authors need more evidence to support the conclusion about “redistribution to secondary lymphoid organs or to tissues”.
Reply: We agree with the reviewer and the sentence has been removed.
- In Fig3. g, the dash line between 0.6 and 0.8 instead of between 0.4 and 0.6.
Reply: The figure has been edited accordingly.
- In Line 284, the sample IDs are different from that in Fig. 6. Please be consistent.
Reply: The sample IDs should now be consistent throughout the manuscript.
- Could the authors do mass spec with the tumor samples from patient 1 to identify the tumor specific antigens? If so, by comparing the sequence of tumor-specific antigens and Igλ chains, the authors may be able to validate that the unique prominently hyperexpanded clonotype CCSYAGSYTFEWVF 306 (IGLV2.11-IGLJ3) was induced by tumor-specific antigens in patient 1.
Reply: We agree that identifying the target antigen is important in understanding response by immune cells and at the same time help in development of potential diagnostic and therapeutic biological products. As mentioned in lines 375-376, we have started patient inclusions for assessing protein expression in tumor material. The study is planned as a prospective study.
Below are some questions that the authors need to address:
- Please define abbreviations on first use in the manuscripts. For example, “PBMC” stands for “peripheral blood mononuclear cell”.
Reply: Abbreviations have now been explained in the text according to appearance.
- In Line 212, it should be “ratio 0.7” instead of “ratio 0,7”.
Reply: Decimal has been inserted instead of comma.
- In Lines 218-220, the authors need more evidence to support the conclusion about “redistribution to secondary lymphoid organs or to tissues”.
Reply: We agree with the reviewer and the sentence has been removed.
- In Fig3. g, the dash line between 0.6 and 0.8 instead of between 0.4 and 0.6.
Reply: The figure has been edited accordingly.
- In Line 284, the sample IDs are different from that in Fig. 6. Please be consistent.
Reply: The sample IDs should now be consistent throughout the manuscript.
- Could the authors do mass spec with the tumor samples from patient 1 to identify the tumor specific antigens? If so, by comparing the sequence of tumor-specific antigens and Igλ chains, the authors may be able to validate that the unique prominently hyperexpanded clonotype CCSYAGSYTFEWVF 306 (IGLV2.11-IGLJ3) was induced by tumor-specific antigens in patient 1.
Reply: We agree that identifying the target antigen is important in understanding response by immune cells and at the same time help in development of potential diagnostic and therapeutic biological products. As mentioned in lines 375-376, we have started patient inclusions for assessing protein expression in tumor material. The study is planned as a prospective study.
Below are some questions that the authors need to address:
- Please define abbreviations on first use in the manuscripts. For example, “PBMC” stands for “peripheral blood mononuclear cell”.
Reply: Abbreviations have now been explained in the text according to appearance.
- In Line 212, it should be “ratio 0.7” instead of “ratio 0,7”.
Reply: Decimal has been inserted instead of comma.
- In Lines 218-220, the authors need more evidence to support the conclusion about “redistribution to secondary lymphoid organs or to tissues”.
Reply: We agree with the reviewer and the sentence has been removed.
- In Fig3. g, the dash line between 0.6 and 0.8 instead of between 0.4 and 0.6.
Reply: The figure has been edited accordingly.
- In Line 284, the sample IDs are different from that in Fig. 6. Please be consistent.
Reply: The sample IDs should now be consistent throughout the manuscript.
- Could the authors do mass spec with the tumor samples from patient 1 to identify the tumor specific antigens? If so, by comparing the sequence of tumor-specific antigens and Igλ chains, the authors may be able to validate that the unique prominently hyperexpanded clonotype CCSYAGSYTFEWVF 306 (IGLV2.11-IGLJ3) was induced by tumor-specific antigens in patient 1.
Reply: We agree that identifying the target antigen is important in understanding response by immune cells and at the same time help in development of potential diagnostic and therapeutic biological products. As mentioned in lines 375-376, we have started patient inclusions for assessing protein expression in tumor material. The study is planned as a prospective study.
Below are some questions that the authors need to address:
- Please define abbreviations on first use in the manuscripts. For example, “PBMC” stands for “peripheral blood mononuclear cell”.
Reply: Abbreviations have now been explained in the text according to appearance.
- In Line 212, it should be “ratio 0.7” instead of “ratio 0,7”.
Reply: Decimal has been inserted instead of comma.
- In Lines 218-220, the authors need more evidence to support the conclusion about “redistribution to secondary lymphoid organs or to tissues”.
Reply: We agree with the reviewer and the sentence has been removed.
- In Fig3. g, the dash line between 0.6 and 0.8 instead of between 0.4 and 0.6.
Reply: The figure has been edited accordingly.
- In Line 284, the sample IDs are different from that in Fig. 6. Please be consistent.
Reply: The sample IDs should now be consistent throughout the manuscript.
- Could the authors do mass spec with the tumor samples from patient 1 to identify the tumor specific antigens? If so, by comparing the sequence of tumor-specific antigens and Igλ chains, the authors may be able to validate that the unique prominently hyperexpanded clonotype CCSYAGSYTFEWVF 306 (IGLV2.11-IGLJ3) was induced by tumor-specific antigens in patient 1.
Reply: We agree that identifying the target antigen is important in understanding response by immune cells and at the same time help in development of potential diagnostic and therapeutic biological products. As mentioned in lines 375-376, we have started patient inclusions for assessing protein expression in tumor material. The study is planned as a prospective study.
Reviewer 2 Report
The present study regarding patients with prostate cancer of different stages strongly indicates tumor-specific T cell-responses from activated B cells suggesting an important role for B cells in the protection against tumors. It is on the whole well written with an interesting message. However, the manuscript suffers from a lack of a broader discussion about the significance of the presented findings taking into consideration the pioneer work of Raposo et al., who reported that B cells secrete antigen presenting vesicles (J Exp Med 183(1996)1161-1172). Hence, extracellular vesicles (EVs) provide a critical means of bidirectional inter-cellular communication rendering immunomodulatory effects by interactions and crosstalks between B cells and T cells as well as between immune cells and tumor cells.
Minor points: lines 51 and 338: adverbs (not adjectives!); line 54: maybe (one word); Table 1: PSA and age are not dimensionless: please fill in! ;line 130: LNs: a new abbreviation: does it refer to SN or N-SN or (SN+N-SN)? ; lines 207, 208, 210: transposition of letters! (PMBC); lines 368,369: What is "prostate cancer antigen", please explain or do they mean PSA alternatively PSMA? ;line 393: what is "tumor antigen specific BCR", please explain!
Author Response
The present study regarding patients with prostate cancer of different stages strongly indicates tumor-specific T cell-responses from activated B cells suggesting an important role for B cells in the protection against tumors. It is on the whole well written with an interesting message. However, the manuscript suffers from a lack of a broader discussion about the significance of the presented findings taking into consideration the pioneer work of Raposo et al., who reported that B cells secrete antigen presenting vesicles (J Exp Med 183 (1996) 1161–1172). Hence, extracellular vesicles (EVs) provide a critical means of bidirectional inter-cellular communication rendering immunomodulatory effects by interactions and crosstalks between B cells and T cells as well as between immune cells and tumor cells.
Reply: We have now included the pioneering work of Raposo et al in our disscussion.
- Minor points: lines 51 and 338: adverbs (not adjectives!);
Reply: Appropriate changes have been made.
- Line 54: maybe (one word).
Reply: It should state now as ´maybe´.
- Table 1: PSA and age are not dimensionless: please fill in!
Reply: Appropriate units have been added.
- Line 130: LNs: a new abbreviation: does it refer to SN or N-SN or (SN+N-SN)?
Reply: The abbreviation is already explained in line 82. Here we explain lymph nodes in general encompassing both Sentinel (SN) and Non-sentinel (N-SN) nodes.
- Lines 207, 208, 210: transposition of letters! (PMBC).
Reply: Abbreviation has been corrected throughout the manuscript.
- Lines 368,369: What is "prostate cancer antigen", please explain or do they mean PSA alternatively PSMA?
Reply: Prostate cancer antigen is the unknown antigen against which the B cells are reacting.
- Line 393: what is "tumor antigen specific BCR", please explain!
Reply: Here we refer to the hypermutated BCR of B cells that have reacted to tumor antigen. This allows for efficient uptake of antigens and subsequent presentation through MHC II to T cells.
Round 2
Reviewer 1 Report
My questions are addressed well.